# Development of a Personalized m/eHealth Algorithm for the Resumption of Activities of Daily Life Including Work and Sport after Total and Unicompartmental Knee Arthroplasty: A Multidisciplinary Delphi Study

**DOI:** 10.3390/ijerph17144952

**Published:** 2020-07-09

**Authors:** A. Carlien Straat, Pieter Coenen, Denise J. M. Smit, Gerben Hulsegge, Esther V. A. Bouwsma, Judith A. F. Huirne, Rutger C. van Geenen, Rob P. A. Janssen, Tim A. E. J. Boymans, Gino M. M. J. Kerkhoffs, Johannes R. Anema, P. Paul F. M. Kuijer

**Affiliations:** 1Department of Public and Occupational Health, Amsterdam UMC, Vrije Universiteit Amsterdam, Amsterdam Public Health research institute, 1081 BT Amsterdam, The Netherlands; p.coenen@amsterdamumc.nl (P.C.); d.smit1@amsterdamumc.nl (D.J.M.S.); gerben.hulsegge@tno.nl (G.H.); h.anema@amsterdamumc.nl (J.R.A.); 2Coronel Institute of Occupational Health, Amsterdam UMC, Academic Medical Center, Amsterdam Public Health Research Institute, 1105 AZ Amsterdam, The Netherlands; p.p.kuijer@amsterdamumc.nl; 3The Netherlands Organization for Applied Scientific Research, TNO, Schipholweg 77-89, 2316 ZL Leiden, The Netherlands; 4Department of Obstetrics and Gynaecology, Amsterdam UMC, Vrije Universiteit Amsterdam, 1081 HV Amsterdam, The Netherlands; ev.bouwsma@amsterdamumc.nl (E.V.A.B.); j.huirne@amsterdamumc.nl (J.A.F.H.); 5Department of Orthopaedic Surgery, Amphia Hospital, Foundation FORCE (Foundation for Orthopaedic Research Care and Education), 4818 CK Breda, The Netherlands; rvangeenen@amphia.nl; 6Department of Orthopaedic Surgery and Traumatology, Maxima Medical Center, 5631 BM Veldhoven, The Netherlands; r.janssen@mmc.nl; 7Chair Value-Based Health Care, Faculty of Paramedical Sciences, Fontys University of Applied Sciences, 5612 AR Eindhoven, The Netherlands; 8Orthopaedic Biomechanics, Department of Biomedical Engineering, Eindhoven University of Technology, 5612 AZ Eindhoven, The Netherlands; 9Department of Orthopaedics, Maastricht University Medical Center, 6229 HX Maastricht, The Netherlands; t.boymans@mumc.nl; 10Department of Orthopaedic Surgery, Amsterdam UMC, Academic Medical Center, 1105 AZ Amsterdam, The Netherlands; g.m.kerkhoffs@amsterdamumc.nl

**Keywords:** knee replacement, convalescence recommendations, patient-tailored advice, return to work, return to sports, eHealth, m/eHealth program

## Abstract

Evidence for recommendations concerning the resumption of activities of daily life, including work and sport, after knee arthroplasty is lacking. Therefore, recommendations vary considerably between hospitals and healthcare professionals. We aimed to obtain multidisciplinary consensus for such recommendations. Using a Delphi procedure, we strived to reach consensus among a multidisciplinary expert panel of six orthopaedic surgeons, three physical therapists, five occupational physicians and one physician assistant on recommendations regarding the resumption of 27 activities of daily life. The Delphi procedure involved three online questionnaire rounds and one face-to-face consensus meeting. In each of these four rounds, experts independently decided at what time daily life activities could feasibly and safely be resumed after knee arthroplasty. We distinguished patients with a fast, average and slow recovery. After four Delphi rounds, the expert panel reached consensus for all 27 activities. For example, experts agreed that total knee arthroplasty patients with a fast recovery could resume cycling six weeks after the surgery, while those with an average and slow recovery could resume this activity after nine and twelve weeks, respectively. The consensus recommendations will subsequently be integrated into an algorithm of a personalized m/eHealth portal to enhance recovery among knee arthroplasty patients.

## 1. Introduction

Knee osteoarthritis is one of the leading causes of pain and disability among adults in many developed countries [1,2,3,4]. In adults 55 years and older, knee osteoarthritis affects approximately 10% of the population, of whom one quarter is severely disabled [5]. In the coming years, the ageing population and the obesity epidemic will likely further increase the number of knee osteoarthritis patients [6]. Partial or unicompartmental knee arthroplasty (UKA) and total knee arthroplasty (TKA) are well-established treatment options for end-stage knee osteoarthritis [7]. If the rising osteoarthritis trend continues, the burden of knee arthroplasty will increase substantially in the coming decade. For example, in the United States, a growth of 673% for total knee arthroplasties from 2005 to 2030 is predicted [8,9]. In the Netherlands, an expected growth of 297% is expected from 2005 to 2030, resulting in 57,900 knee arthroplasties each year [10]. The highest increase in knee arthroplasty is expected among patients of working-age, i.e., below the age of 65. These relatively young patients are still active in work and sports activities, and tend to have high preoperative expectations concerning return to daily life activities after surgery [11].

Despite the good clinical outcomes with significant pain relief and improved knee function in most patients who receive knee arthroplasty, approximately 20% of the patients are dissatisfied postoperatively [12]. In a recent study, Mahdi et al. concluded that 6–30% of knee arthroplasty patients had unfulfilled expectations regarding their ability to resume daily activities after surgery [13]. From a study with interviews among 45 knee arthroplasty patients, it was concluded that patients received little guidance and support regarding the resumption of daily activities [14]. Since the strongest predictor of patient satisfaction appears to be the fulfilment of preoperative expectations [15,16], it is likely that setting realistic recovery goals regarding the resumption of daily activities would lead to improved satisfaction. Unfortunately, in many countries, such as the Netherlands, guidelines for healthcare professionals provide no clear advice regarding recovery recommendations for daily activities, including work and sport, after knee arthroplasty [17]. Probably, the reason is that recommendations are often based on expert opinions of healthcare professionals, as scientific evidence is limited. Therefore, patients receive no advice at all, or conflicting advice. Well-defined and multidisciplinary recovery recommendations for the resumption of daily life activities are needed to set realistic recovery expectations.

To make sure advice regarding the resumption of daily life activities is and remains relevant for all, the advice should be personalized to the patient’s needs. For example, the pace of recovery could differ substantially between patients, of which a hypothetical example can be seen in Figure 1. Many patients have their ups and downs during their own recovery process, as a result of which advice should be adapted to the changing patient’s needs. For that reason, providing personalized and adaptive guidance is of utmost importance.

A recent review demonstrated that the resumption of activities of daily life after knee arthroplasty can be improved with the help of integrated care programs, such as those provided by m/eHealth [18]. m/eHealth has the possibility to provide personalized and adaptive guidance to patients. Moreover, due to the importance of efficient use of hospital resources and the cuts of hospitalization costs, m/eHealth programs can support and guide patients during the recovery period at home. With m/eHealth programs, efficient use of hospital resources can be maintained and the needs of the patients can be fulfilled [19,20,21,22,23]. However, before such an m/eHealth program can be developed, consensus among different healthcare professionals about recommendations for return to daily life activities needs to be reached. For personalized advice it is further necessary to distinguish different recovery rates.

We aimed to conduct a Delphi study to yield multidisciplinary consensus on the timing at which patients can return to activities of daily life, including work and sport, after knee arthroplasty. To secure personalized and adaptive recovery recommendations, we developed recommendations for three patient groups for UKA and TKA: patients with a fast recovery, an average recovery and a slow recovery. The multidisciplinary recovery recommendations developed will form the basis of algorithms that will be implemented in a personalized m/eHealth portal for knee arthroplasty patients.

## 2. Materials and Methods

### 2.1. Study Design

A Delphi procedure was used to develop multidisciplinary recovery recommendations for the resumption of daily life activities, including work and sport. The Delphi method is used to achieve consensus when the literature is inconclusive or incomplete [24], and previous studies have demonstrated that this method is effective in achieving consensus on when to resume activities of daily life after surgical procedures [25,26]. An overview of the study design is depicted in Figure 2. First, a literature search was performed in order to review existing evidence on the recovery and recovery recommendations after UKA and TKA. Relevant convalescence activities were selected and a questionnaire was developed. Then, experts were recruited and in three consecutive online rounds they were asked to formulate detailed recovery recommendations on the timing of gradual (i.e., stepwise) resumption of the selected activities after UKA and TKA. Additionally, one face-to-face consensus meeting was organized. Data collection took place between May and September 2019. The recommendations obtained from this Delphi procedure will be implemented as algorithms in a personalized m/eHealth portal.

### 2.2. Recruitment of Expert Panel

Fifteen experts participated in this Delphi study: six orthopedic surgeons, three physical therapists, five occupational physicians and one physician assistant. Twelve experts were male, three female. Work experience varied from two to 25 years, and experts treated on average nine knee arthroplasty patients per month. We recruited experts from fifteen different hospitals and practices throughout the Netherlands and thereby assured sufficient expertise in knee arthroplasty care from different perspectives. Experts were recruited using the authors’ network. One expert dropped out after the first Delphi round. Data of this expert were only used during analysis of the first questionnaire round. The other fourteen experts completed the full protocol.

### 2.3. Review of Existing Recovery Recommendations after UKA and TKA

A literature search was carried out in the electronic scientific database PubMed from which current evidence regarding recovery and return to normal activities was reviewed. Search terms included free words in the title or abstract, and mesh terms depicting “knee arthroplasty”, “recovery”, “convalescence”, “rehabilitation”, “return to activities”, “return to work”, “return to sports”. Articles were assessed for eligibility, including studies reporting return to daily activities or return to work as primary or secondary outcome measures. Articles other than randomized controlled trials, systematic reviews or international guidelines were excluded. In total, 21 articles were included. From the included articles, relevant activities of daily life including work and sport and information about the timing of the resumption of those activities after surgery were collected. Additionally, recovery recommendations of 39 Dutch hospitals and clinics from flyers, brochures and websites were examined. With these 39 hospitals and clinics, we identified the recovery recommendations provided to at least 70% of all knee arthroplasty patients in the Netherlands. The summarized recovery recommendations and the review of the literature were used as guidance for the expert panel and for the development of the questionnaire.

### 2.4. Selection of Relevant Activities and Development of Delphi Questionnaire

The following questionnaires were used to select additional relevant activities of daily life: the Functional Ability List (FAL), PROMIS-Physical Functioning v2.0 (PROMIS-PF) and the Work, Osteoarthritis or Joint-Replacement Questionnaire (WORQ). The FAL contains 59 different activities that represent general functional abilities and can be used to assess functional ability in daily life [27]. The PROMIS-PF examines physical function and includes 121 activities covering simple everyday activities, as well as more complex activities that require a combination of skills [28]. The WORQ contains 13 activities that are often performed at the workplace [29,30]. Based on these three questionnaires and the literature identified (mentioned under Section 2.3), during a meeting the research team selected activities that they considered relevant for knee arthroplasty rehabilitation. Based on consensus, 27 items were selected as most relevant (Appendix A). In addition, the expert panel was asked to examine if activities were missing and/or if activities were less relevant. Thereafter, for the selected 27 activities, the research team predicted the recovery length of each specific activity in order to develop the corresponding questionnaire timeline, i.e., after how many postoperative days, weeks or months can a patient gradually resume this activity.

To secure personalized and adaptive recovery recommendations, the aforementioned questionnaire was formulated for UKA and TKA surgeries and three patient groups: patients with a fast, average and slow recovery for return to daily activities.

### 2.5. Delphi Protocol

Before the first Delphi round, the expert panel received the summary of the recovery recommendations of the 39 Dutch hospitals and clinics and the reviewed literature. This summary was to be used as guidance. During the first questionnaire round, experts were asked to independently score if, and to what extent, an activity could be resumed after knee arthroplasty. An example of the activity “walking without walking aids” is presented in Table 1. At each time point (i.e., day 1 after surgery, or week 8 after surgery), experts scored an ability score for each activity: 3 (very limited), 2 (limited), 1 (slightly limited) or 0 (normal). These ability scores are part of the original Dutch FAL [27] and the research team prespecified the scores for knee arthroplasty rehabilitation. The ability scores varied from two to four categories. Additionally, the time line varied from 8 weeks until 24 months after surgery.

For each Delphi round, the mode and median values of each ability score and the mean consensus per activity were calculated. Consensus was reached when at least two thirds (66.7%) of the experts agreed with the ability score on all time points for activities with three or four categories. For activities with two categories, consensus was reached when at least 75% of the experts agreed with the ability score on all time points.

After all experts completed the first questionnaire, the research team revised the questionnaire. The mode and median values from the first round were graphically presented to the expert group in the second questionnaire. For time points with consensus, the consented ability score was depicted in the questionnaire; experts only had to rate the ability score for the time points without consensus during the second Delphi round. This procedure was repeated for the third Delphi round. When experts decided that activities could not be fully resumed after the last time point (i.e., a ceiling effect), extra time points (additional postsurgical weeks or months) were added during this third round. Moreover, in the third questionnaire and for time points at which no consensus was yet reached, the mode was depicted on the questionnaire if the consensus was above 50% to help the experts choose their ability score.

The last and fourth Delphi round was a consensus meeting in which nine of the fourteen experts met face-to-face: three occupational physicians, two physical therapists, two orthopaedic surgeons and one physician assistant. The five other experts were unable to attend the meeting. During this meeting, experts were able to discuss the activities and time points at which no consensus was yet reached. At the beginning of the meeting, the results of the previous Delphi questionnaires were presented to the experts. During the group discussion, the nominal group technique was used to ensure equal participation of all experts [24]. The nominal group technique is a structured method for a group discussion where you encourage contribution from everyone. Experts were asked to complete the final Delphi questionnaire, taking into consideration the insights and arguments conveyed.

After the consensus meeting, the results of the final Delphi questionnaire were sent to the expert panel, and experts were asked to give their approval for the final set of multidisciplinary recommendations for the gradual resumption of activities after knee arthroplasty.

## 3. Results

### 3.1. Relevant Activities

None of the experts reported missing or irrelevant activities in the questionnaire. The experts asked to change the specifications for the ability scores for the activities “kneeling” and “crouching”, as they argued that the most demanding score of 0, i.e., normal (kneeling for 60 or more minutes and crouching for 60 or more minutes), would be too demanding after surgery. The expert panel discussed the activities “jogging” and “knee demanding sports activities” during the consensus meeting and eventually agreed that running a short distance (e.g., to catch a train) can be recommended. However, jogging five kilometers or more and knee demanding sports activities such as basketball or soccer are not to be recommended to resume at all after knee arthroplasty.

### 3.2. Consensus

Appendix A shows the percentage of consensus reached for each activity, across time points, per Delphi round for TKA patients with an average recovery rate (other categories not reported). It can be seen that the mean consensus was reached in many activities in round one. However, all activities contained individual time points at which no consensus was yet reached.

In round one, experts did not reach overall consensus on any activity. In round two, experts reached consensus on three activities. In round three, experts reached consensus on another two activities. In round four, the experts reached consensus on the other 22 activities. To illustrate, Figure 3 represents the mean percentage of consensus for six activities: (A) daily life, showing walking and household activities; (B) sports, showing cycling and jogging short distances; and (C) work, showing working for 4 and 8 h per day. The duration until consensus was reached was comparable for activities of daily life, work and sports, after both UKA and TKA. Extra time points were added in round three because the experts judged that several activities could not be fully resumed at the last presented time point. As a consequence, the mean consensus in round two and three were comparable. For household activities (UKA and TKA), cycling (UKA), working for 4 h (UKA) and working for 8 h (UKA), consensus was reached after the third Delphi round. In the fourth Delphi round (i.e., consensus meeting) consensus was reached for all the remaining activities.

### 3.3. Recovery Recommendations

For TKA patients with an average recovery, examples of activities that could be resumed within two months after surgery are prolonged sitting (after 3 days), taking a shower (after 3 days) and prolonged standing (after 3 weeks). Examples of activities that could be resumed after two months or more are knee-demanding activities, such as climbing and/or clambering (after 11 weeks), kneeling (after 4 months) and crouching (after 4 months).

There was a difference in recovery recommendations between UKA and TKA for several activities. For patients that recover fast, experts agreed that UKA patients could resume 4 out of the 27 activities, such as crouching, sooner than TKA patients. For patients that recover on average, this was the case for 12 out of the 27 activities, such as pushing and pulling. For the patients that recover slow, UKA patients could resume 8 out of the 27 activities earlier, such as cycling. Additionally, for all three groups of patients, experts agreed that UKA patients could return to work sooner, as compared to TKA patients with an average recovery, namely, after 5 versus 6 weeks for light knee-demanding work, after 11 weeks versus 16 weeks for moderate knee-demanding work, and after 6 versus 12 months for heavy knee-demanding work.

As an illustration, Table 2 represents the recovery recommendations for the activity “cycling” for all six groups, i.e., three recovery rates for UKA and TKA patients.

## 4. Discussion

### 4.1. Main Findings

After three questionnaire rounds and one consensus meeting, consensus was reached on detailed recovery recommendations regarding the resumption of 27 activities of daily life following UKA and TKA. For TKA patients with an average recovery rate, examples of activities that can be resumed relatively fast after surgery are sitting (after 3 days) and standing (after 3 weeks). Examples of activities that can be resumed relatively late after surgery are crouching (9 weeks) and kneeling (10 weeks).

### 4.2. Comparison with Other Studies

To our knowledge, no uniform and multidisciplinary recommendations on the resumption of daily life activities after knee arthroplasty surgery exist. Particularly given the rapid increase in knee arthroplasty, it is remarkable that these recommendations are still lacking. In a study performed by our own research group, in which we reviewed recovery recommendations in usual care for the resumption of daily activities and work after knee arthroplasty in 39 Dutch hospitals, we demonstrated that these recommendations are often lacking and vary considerably between hospitals [31]. Internationally, several studies investigated guidelines and (exercise) recommendations for postacute, postoperative physical therapy after UKA or TKA, but none of these studies provided (evidence on the) recovery recommendations for activities of daily life [32,33,34,35]. These findings are in line with the studies from Nouri et al. [14] and Mahdi et al. [13], who both showed that patients expressed the need for mutual guidelines and advice regarding return to daily activities. Given the finding that preoperative expectations are one of the main determinants to influence postoperative patient satisfaction, realistic and multidisciplinary recovery recommendations regarding resumption of daily life activities could improve patient satisfaction after surgery [19,23,36,37].

Several Delphi studies were conducted in the field of knee arthroplasty rehabilitation. Westby, Brittain and Backman (2014) used the Delphi method to obtain best practice recommendations for TKA rehabilitation [38]. The authors concluded, amongst other things, that supervised rehabilitation interventions and short-term follow-up care in the first two years after surgery is recommended for improving the quality of the rehabilitation [38]. A second Delphi study by Plenge et al. examined factors that could optimize perioperative care for knee arthroplasty patients and patient outcomes after knee arthroplasty [39]. This Delphi study concluded that poor general health and impaired cardiovascular functional status are one of the most important determinants for poor patient outcomes after surgery, including postoperative pain and immobilisation. Additionally, multidisciplinary planning was considered as one of the most important determinants to improve patient reported outcomes following primary knee arthroplasty. Lastly, quality indicators for knee arthroplasty rehabilitation were examined by Westby, Marshall and Jones (2018) [40]. This study reached consensus on 36 quality indicators for total knee arthroplasty. The experts agreed that multidisciplinary preoperative education should be performed, addressing at least the surgical procedure, risks and benefits, patient expectations regarding the (outcome of the) surgery, pain management strategies, home preparation, assistive devices and postoperative care and rehabilitation. These three Delphi studies all provided necessary elements for enhancing perioperative care of knee arthroplasty patients and helped improve the quality of postoperative knee arthroplasty rehabilitation. However, when compared to the current Delphi study, none of the previous studies focused on recommendations for the resumption of daily activities after knee arthroplasty. Our Delphi study addresses this gap in the literature and provides information for the development of best-practice recommendations and guidelines based on consensus of relevant experts. The developed recommendations are also important for discussing preoperative patient expectations and could therefore, for example, be combined with the preoperative education program as described by Westby, Marshall and Jones (2018) [40].

The three Delphi studies mentioned in the previous paragraph reached consensus in three or four Delphi rounds [38,39,40]. In line with these studies and an earlier Delphi study from our research group for the development of recovery recommendations after abdominal surgery, we reached consensus after four Delphi rounds [26]. Considering the variation in recovery advice between hospitals and healthcare professionals [31], we consider this relatively fast. A difference between the two Delphi studies from our research group, however, was the timing of the group discussion, which our colleagues did after the first Delphi round [26]. Having a group discussion at the beginning of the study provides the opportunity to address important or possible unclear aspects of the questionnaire and discuss insights and reasons for specific ability scores. However, we only had the opportunity to do the group discussion at the end. A group discussion in an earlier phase of the study might have been more efficient to discuss activities with large variations in ability scores. Nonetheless, as we already reached consensus for some activities during the first three questionnaire rounds, this group discussion at the end gave us the opportunity to take sufficient time to discuss the activities and time points on which experts did not yet agree to reach consensus on all remaining activities.

### 4.3. Strengths and Limitations

A major strength of this study is the Delphi design, incorporating experts involved in different domains of knee arthroplasty rehabilitation. The heterogeneity of the research team and expert panel provided us with insights from different perspectives from the most relevant disciplines, such as orthopaedics, physical therapy and occupational medicine. Only one expert dropped out during the questionnaire rounds and fourteen experts completed the entire study. Experts were asked to complete the questionnaires anonymously, which avoided domination by one or more experts. Dominance and the influence of other experts was also prevented by the use of the nominal group technique during the consensus meeting. This consensus meeting gave us the opportunity to collectively reflect on the activities and developed recommendations, and, if needed, to add or change categories. However, a single face-to-face consensus meeting for discussing and conveying arguments when not all experts were present can be considered as a limitation. We did, however, ensure that at least two experts from all professions and disciplines were present at that meeting, and provided all absent experts with an extensive summary and possibility to react and reflect on the decisions made after full consensus for all 27 activities.

In the current Delphi study, our expert panel agreed on return to daily activities for patients with three recovery rates (i.e., fast, average and slow recovery). Experts were asked to self-determine what they considered to be fast, average and slow. To our knowledge, this is the first Delphi study to distinguish between patient groups for the same surgical procedure, making it possible to provide comprehensive personalized and adaptive advice. These recommendations will be used as algorithms that will be implemented in an m/eHealth program for knee arthroplasty patients.

The 27 activities of the Delphi questionnaire were selected by the research team using the flyers and brochures from Dutch hospitals, and FAL, PROMIS-PF and WORQ as guidance. To date, there is no single questionnaire or instrument that determines relevant activities for postoperative rehabilitation, and it can be questioned whether we missed relevant activities. We based our selection on activities that are knee-demanding, and decided to exclude activities such as “concentrating”, “reading” and “moving your head”. All fourteen experts agreed with the selected activities and did not report missing activities or judged one or more of these 27 activities as less relevant.

No patients participated in our expert panel. Patients can be considered the real experts regarding the resumption of activities after surgery. Yet, a study by Bouwsma et al. showed that patients sometimes underestimate their ability to resume activities, including work after surgery [41]. Additionally, experts treat several patients per week or per month, whereas patients only experience their own recovery. Therefore, we chose not to include patients in our expert panel. However, it must also be said that experts only see patients for few moments during the rehabilitation, and future research is needed to evaluate if the developed recovery recommendations are realistic in daily life.

Finally, our algorithm takes different recovery rates into account, but does not differentiate between other relevant factors that could influence the recovery, such as personal, clinical and occupational factors [41,42]. For more advanced personalized advice, these factors should be incorporated in future recovery recommendations for daily life activities. However, the prospective collection of detailed recovery data of actual patients would be required first. Additionally, generalizing the developed recommendations should be done with caution, as recovery trajectories and rehabilitation programs after knee arthroplasty may differ between countries and populations.

### 4.4. Algorithm Development for Future Studies

Using the set of multidisciplinary recovery recommendations from our Delphi study, we are currently developing a personalized m/eHealth program for knee arthroplasty patients. Using this m/eHealth program, we will collect detailed recovery data of knee arthroplasty patients to evaluate the validity of our developed recovery recommendations.

Our research group has previously shown that personalized m/eHealth programs are effective for return to daily life activities, including work, after gynecological and abdominal surgery [19,23,36]. In our m/eHealth portal for knee arthroplasty patients, we will transform the recovery recommendations into two algorithms, one each for UKA and TKA.

In the algorithm, patients will be categorized into one of the three recovery trajectories (i.e., fast, average or slow), based on their own preoperative recovery expectations regarding return to work. Patients will only receive advice about activities they select to be important in their daily life. Additionally, we will expand the algorithm with more complex activities. We will combine several recovery recommendations for single activities, such as walking, standing or carrying, that are needed for complex activities such as grocery shopping. In our m/eHealth portal, patients can choose the degree of these single activities (i.e., carrying 5 kg or carrying 15 kg) for their complex activities. Patients will be asked weekly which of their selected activities they have resumed after surgery. This information will be used to adapt the patient’s advice if needed (i.e., shifting to a faster or slower recovery rate group). This innovative algorithm will allow the m/eHealth portal to provide personalized and adaptive recovery recommendations.

The (cost) effectiveness of our m/eHealth portal on the postoperative recovery and duration of return to daily life activities after knee arthroplasty will be evaluated in a randomized controlled trial in ten Dutch hospitals and clinics. This trial is registered in the Netherlands Trial Register, number NL8525. The results from this study will provide healthcare providers and policy makers with guidance to improve knee arthroplasty patients’ care and possible future implementation of m/eHealth in Dutch orthopaedic hospitals and clinics. Multidisciplinary recovery recommendations for return to daily life activities, including work and sports, could bridge the gap between recovery expectations and actual recovery time. With a faster return to daily life activities after knee arthroplasty, patients are likely to experience an improvement in their quality of life [43]. Additionally, this will potentially benefit employers and society as a whole by increasing societal participation, therefore reducing costs due to productivity loss and sick leave.

## 5. Conclusions

A multidisciplinary expert panel of six orthopaedic surgeons, three physical therapists, five occupational physicians and one physician assistant achieved full consensus on recovery recommendations regarding the gradual resumption of 27 daily activities after unicompartmental and total knee arthroplasty. The final set of recovery recommendations will be transformed into two algorithms that will be integrated into a personalized m/eHealth program, providing knee arthroplasty patients with patient-tailored recovery advice.

## Figures and Tables

**Figure 1 ijerph-17-04952-f001:**
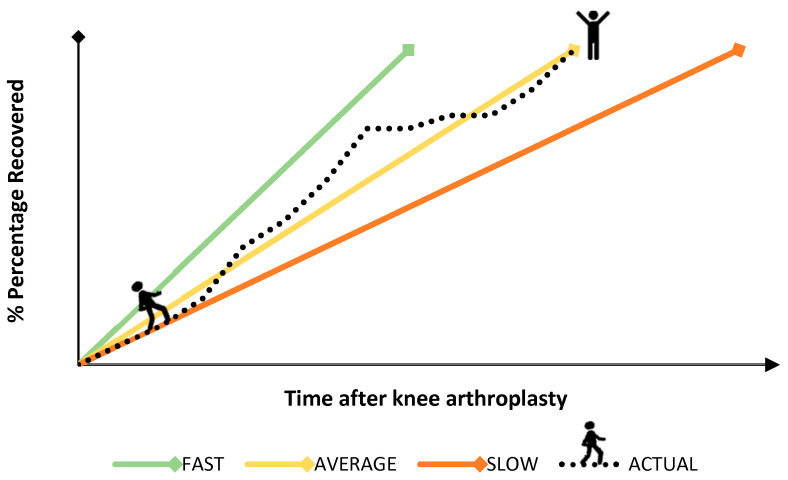
Hypothetical and schematic representation of three potential trajectories of recovery (i.e., fast, moderate and slow) for return to daily activities after knee arthroplasty. An example of a realistic and actual recovery trajectory of a patient, showing ups and downs during the recovery process, is also shown.

**Figure 2 ijerph-17-04952-f002:**
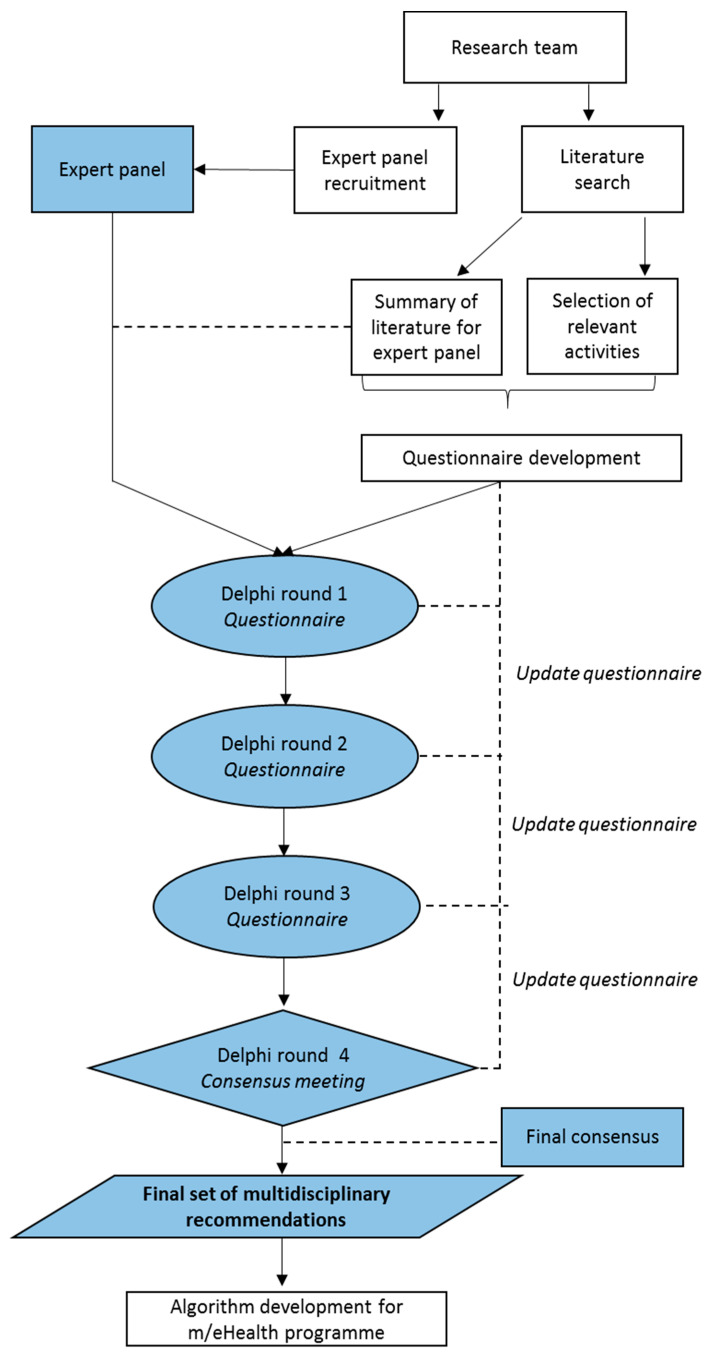
Flow diagram depicting the Delphi study protocol.

**Figure 3 ijerph-17-04952-f003:**
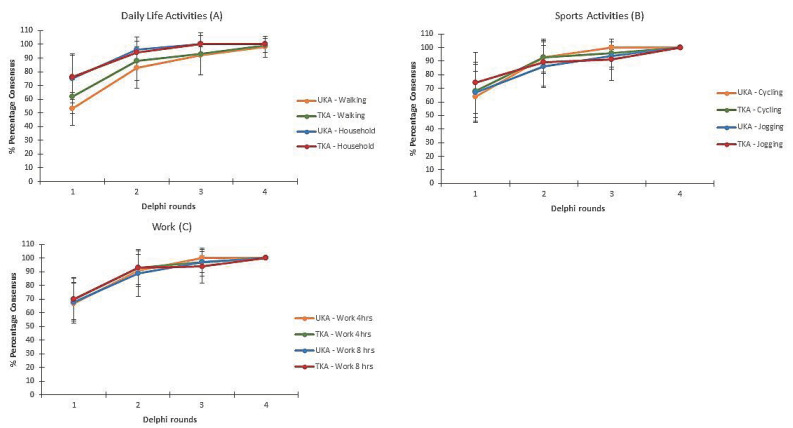
Mean percentage of consensus per Delphi round for patients with an average recovery rate for three categories of activities: daily life (**A**), sports (**B**) and work (**C**). The error bars represent the standard deviation.

**Table 1 ijerph-17-04952-t001:** Example of the Delphi panel questionnaire, in this case, for the activity “Walking without walking aids”.

	1 day	2 days	3 days	4 days	5 days	6 days	1 wk	2 wks	3 wks	4 wks	5 wks	6 wks	7 wks	8 wks
UKA (F)														
UKA (A)														
UKA (S)														
TKA (F)														
TKA (A)														
TKA (S)														
**Ability score** **0:** Normal, can walk at least 2 h without walking aids **1:** Slightly limited, can walk 1 h (maximum) without walking aids **2:** Limited, can walk in and around the house without walking aids **3:** Very limited, cannot walk without walking aids														

UKA = unicompartmental knee arthroplasty; TKA = total knee arthroplasty; 2 days = two days after surgery, etc.; 1 wk = one week after surgery, etc.; (F) = fast recovering patients; (A) = average recovering patients; (S) = slow recovering patients.

**Table 2 ijerph-17-04952-t002:** Recovery recommendations for “cycling” for all six groups, i.e., three recovery rates for UKA and TKA.

	1–3 wks	4 wks	5 wks	6 wks	7 wks	8 wks	9 wks	10 wks	11 wks	12 wks	14 wks	16 wks	20 wks		
**UKA (F)**															
**UKA (A)**															**Not allowed**
**UKA (S)**															**2–3 km**
**TKA (F)**															**3–10 km**
**TKA (A)**															**30–40 km (N)**
**TKA (S)**															

Note: wk(s) = weeks; (N) = patients can perform and resume the activity normally, i.e., without limitations; (F) = fast recovering patients; (A) = average recovering patients; (S) = slow recovering patient.

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
