# Peer review of "Development of a Personalized m/eHealth Algorithm for the Resumption of Activities of Daily Life Including Work and Sport after Total and Unicompartmental Knee Arthroplasty: A Multidisciplinary Delphi Study"

_ijerph, 2020, doi:10.3390/ijerph17144952_

Round 1
Reviewer 1 Report
This is an interesting study investigation the rehabilitation following TKR to enable the setting of guidelines on expected achievement for three sets of patients (fast,average and slow). The Delphi methodology was appropriate although I question only one face-to-face interview and the low turn out or expert surgeons at that meeting (this should be emphasised as a major limitation)
My main issue with this type of research is that it by necessity has to be country/population dependent. By this I mean different societies have different expectations/demands and these cannot be labelled under one "algorhythm" eg the researcher's excluded prolonged kneeing and crouching from the expectation of TKR but in some cultures this is the overriding requirement to ensure a satisfactory recovery (Asian countries).
I think it is satisfactory to have these guidelines for the Netherlands but they cannot be expanded to other countries necessarily. That should be listed as a significant limitation.
Author Response
The authors would like to thank the reviewer for taking the time to assess our manuscript and for the constructive feedback provided. Please see the attachment for a point by point response. Changes to the manuscript are highlighted with track-changes and can be found in our revised document.

Reviewer 2 Report
Dear Authors:
Thank you for submitting the manuscript entitled "Development of a Personalized m/eHealth Algorithm for the Resumption of Activities of Daily Life including Work and Sport after Total and Unicompartmental Knee Arthroplasty: a Multidisciplinary Delphi Study". It is an interesting topic. The reviewer appreciates the authors' attempt to reach a multidisciplinary consensus on the timing at which patients can return to activities of daily life including work and sport after UKA and TKA. I have the following detailed comments.
1. Introduction
- Background and controversy are adequately introduced.
- The purpose is appropriate.
2. Methods
- Since this study include a literature review, please add what is the inclusion and exclusion, how many articles are included?
- Any reference for the ability score with a 0-3 scale?
- My concern here is that the method is more of an agreement of discussion. Any validation of actual patients with follow-ups?
3. Results
- Again, the short and long term recovery recommendations are based on the agreement among experts. The definitions for "short" and "long" term seems very objective judgment. It looks like to me that the experts were actually allocating the time when the patients were able to perform certain activities to define short or long term recovery.
4. Discussion
- Line 310: Please elaborate what were the results of these studies? This section is important as the readers would like to get useful information from this manuscript. Please compare with previous studies and discuss what is new and unique in this study.
- Line 347: what were the definitions for the three recovery rates?
- Line 378: how was the expectation determined? from patient or surgeon?
6. Conclusions
- The conclusion is OK.
7: Reference
- The reference is ok.
My first concern is that although the author described in detail how the questionnaires and discussions were carried out, the results were barely staying at the theoretical level. No actual patients were involved to validate/evaluate the achieved recommendations. Applications of these recommendations on patients are needed, and those results are more meaningful to the readers.
The author mentioned that part of the guidelines were from a systematic literature review. But the results were missing. It needs inclusion, exclusion, but not briefly stated.
The author defined three recovery rates. But again, these definitions are objective judgments and just based on the agreement among experts.
Author Response
The authors would like to thank the reviewer for taking the time to assess our manuscript and for the constructive feedback provided. In general, we agree with the comments and have revised our manuscript accordingly. Please see the attachment for a point by point response to all comments in red. Changes to the manuscript are highlighted with track-changes and can be found in our revised document.

Reviewer 3 Report
The authors of this manuscript have done an excellent job describing the patterns or TKA literature with day to day questions of functions. Based on models and projections, this Delphi projection models helps the surgeon and the patient have time frames of recovery and goals and platforms. Although, the study presents a platform the patient is robust in the applications in the future to TKA literature
Author Response
The authors would like to thank the reviewer for assessing our manuscript and the very kind words. As mentioned in our discussion, we will also evaluate the validity of these recommendations using our the personalised m/eHealth program including about 400 patients.

Round 2
Reviewer 2 Report
Dear Authors:
Thank you for submitting the revised manuscript.
My concerns and comments about literature review inclusion and exclusion have been addressed.
Additional references have been added.
The Method and Results now provide more details of the study design.
Discussion about the unique aspects of the current study has been added.
Comments from other reviewers have also been addressed.
Nice job!